# Near Infrared Spectroscopy as a Traceability Tool to Monitor Black Soldier Fly Larvae (*Hermetia illucens*) Intended as Animal Feed

Shanmugam Alagappan [1,2], Louwrens C. Hoffman [1,2,3], Sandra M. Olarte Mantilla [1], Deirdre Mikkelsen [1,4], Peter James [5], Olympia Yarger [6] and Daniel Cozzolino [1,*]

1  Centre for Nutrition and Food Sciences, Queensland Alliance for Agriculture and Food Innovation (QAAFI), The University of Queensland, Brisbane, QLD 4072, Australia
2  Fight Food Waste Cooperative Research Centre, Wine Innovation Central Building Level 1, Waite Campus, Urrbrae, SA 5064, Australia
3  Department of Animal Sciences, University of Stellenbosch, Private Bag X1, Matieland, Stellenbosch 7602, South Africa
4  School of Agriculture and Food Sciences, Faculty of Science, University of Queensland, Brisbane, QLD 4072, Australia
5  Centre for Animal Science, Queensland Alliance for Agriculture and Food Innovation (QAAFI), The University of Queensland, Brisbane, QLD 4072, Australia
6  Goterra, 14 Arnott Street, Hume, ACT 2620, Australia
*  Correspondence: d.cozzolino@uq.edu.au

**Abstract:** The demand for animal proteins, especially from pork and poultry, is projected to increase significantly due to rapid growth in population and underlying socio-economic conditions. Livestock rearing using conventional feed ingredients is becoming challenging due to climate change and several other factors, thereby suggesting the need for alternative, viable and sustainable animal feed sources. The use of black soldier fly larvae (BSFL) (*Hermetia illucens*) as a component in animal feed is a promising candidate due to their ability to valorise different organic waste streams. The nutrient composition of BSFL reared on organic waste streams is also comparable to that of several conventional animal feed ingredients and varies depending upon the feed, rearing conditions, and the morphological stage of the larvae. The identification of organic waste is of importance as it can determine not only the composition but also the safety issues of BSFL as an animal feed ingredient. The objective of this study was to determine the ability of near-infrared (NIR) spectroscopy to trace the food waste used to grow BSFL. Samples of BSFL (5th and 6th instar BSFL; *n* = 50) obtained from a commercial production facility were analysed using NIR spectroscopy. Partial least squares discriminant analysis (PLS-DA) was employed to develop the models. The outcomes of this study revealed that NIR spectroscopy could distinguish different larval instars and suggested the importance of larval instars in developing calibration models for traceability applications. The developed PLS-DA model could predict the feed source used for rearing the 5th instar larvae ($R^2$ value: 0.89) and 6th instar pre-pupae ($R^2$ value: 0.91). This suggests that NIR spectroscopy could be used as a non-invasive traceability tool for BSFL and to assist in selecting the suitable time frame for larvae harvesting in commercial facilities.

**Keywords:** black soldier fly larvae; traceability; animal feed; NIR spectroscopy; organic side streams

## 1. Introduction

The global demand for protein obtained from livestock is increasing rapidly because of, amongst other factors, the globally improved income in the upper-middle-class sector. The average demand for livestock-derived protein is observed to increase up to 30 g per day worldwide by the year 2050 [1]. Livestock rearing and production by current practices have an environmental impact as there is a loss of energy and nutrients during the bioconversion

process of plant biomass by animals [2]. Feedstock is observed to be responsible for around 65–85% of livestock production costs. The water footprint, land footprint, and mineral footprint of livestock production using conventional feed ingredients such as grain and cereals have suggested the need for viable and sustainable feed sources [3].

Annually, 1.3 billion tonnes of food produced for human consumption is wasted, and the management of these waste streams through landfills and incineration is observed to have a negative impact on the environment [4]. The valorisation of this food waste using black soldier fly larvae (BSFL) and their subsequent use as feedstock has received much attention lately [5–8].

Black soldier flies (*Hermetia illucens* L. 1758; Diptera: Stratiomyidae) are widely distributed in tropical and subtropical regions [9]. The life cycle of BSFL begins with the adult female fly's oviposition, laying around 200–600 eggs [10,11]. The eggs are laid on dry surfaces, and the hatching of eggs into larvae usually takes place within 4 days [12]. BSFL go through six larval instars, and these larvae are polyphagous saprophagous and can actively forage on a variety of organic products [13]. The duration of this larval stage can range between 10 and 52 days depending upon the type of substrate, temperature, and humidity of the feeding environment [10]. Larval instars 1–4 are whitish in colour, creamy in appearance, and differ in length and size. The length of the first, second, third, and fourth instar larvae vary between 2–5 mm, 6–9 mm, 10–13 mm, and 14–16 mm, respectively. The fifth instar larvae are brownish-grey in colour and are about 17–20 mm in length. The pre-pupae (sixth instars) are characterised by a prominent dark brownish-black colour and are around 20–22 mm in length [14–16]. It is noteworthy that the morphological features of different larval instars are influenced by the substrate and rearing conditions [10]. BSFL, upon reaching the pre-pupal stages, stop feeding and move away from the feed [17]. The emergence of pupa from sixth-instar BSFL or the pre-pupa takes 14 days and involves six stages of moulting [11]. The pupae develop into adult BSF normally in 10–14 days under suitable environmental conditions. The adult BSF are 15–20 mm in length [18]. The adult flies do not have mouthparts, and they rely on their fat stores and liquid supplements for survival and mating [17]. The mating between the male and female adult flies commences two days after eclosion during flight or in the ground under sunlight or similar environmental conditions, and successful mating leads to ovipositing in two days [19,20].

BSFL exhibit the potential to valorise organic side streams such as spent grains, food waste and several other waste products resulting from the agricultural process [6,7]. The nutritional potential of BSFL reared on organic side streams is composed of crude fat varying between 15 and 49% (dry weight basis), and the protein content varies between 30 and 50% (dry weight basis) [21–23]. The protein content in BSFL generally decreases with age [23]; this is attributed to the enzymatic catalysis of protein, resulting in the development of chitin [11]. The amino acid profile of BSFL is constituted of essential and non-essential amino acids [24], with lysine as a predominantly distributed amino acid (~5.6% of the total amino acid composition) [25]. BSFL are also an excellent source of arginine owing to the high bioavailability of this compound in the larvae [26]. Amino acids such as tryptophan, threonine, valine, and cysteine, which are considered important in swine and chicken diets, are also distributed at better proportions in BSFL when compared to soybean meal and other plant-based feeds [26]. It is noteworthy that the amino acid profile of BSFL is not drastically influenced by the amino acid composition of the substrate fed to BSFL [27]. The fatty acid profile of BSFL is dominated by saturated fatty acids that account for 60–80% of the total fatty acid content. The predominant saturated fatty acid in BSFL is Lauric acid (C12:0) (32–60%) followed by palmitic acid (C16:0) (8–20%) and oleic acid (C18:1 *n*-9) (5–12%) [21,24]. The concentration of essential unsaturated fatty acids such as linoleic acid and $\alpha$-linolenic acid in BSFL is influenced by the substrate used for rearing [22]. BSFL also accumulates various minerals such as manganese, calcium, iron, phosphorous, and even certain vitamins from the substrate used for rearing [28,29]. Overall, BSFL reared using organic side streams exhibit the potential to serve as a sustainable feed ingredient. However, the commercialisation of BSFL as a feed ingredient is not widespread due to

the lack of specific regulations pertaining to their use as feed [4]. Real-time information regarding various safety issues (biological, chemical, and microbial) of BSFL and rapid methods to monitor their quality (nutritional and safety) when reared on organic side streams are required so as to establish control points and tolerable limits for different quality parameters of BSFL when reared on different organic side streams [30,31]. This will eventually assist in the development of specific regulations governing their use as feed.

Traceability in the food supply chain is defined as "*the ability to trace a food, feed, or any other substance that's intended to be used either as food or feed*" [32]. It is identified as one of the essential factors that is responsible for ensuring food safety across the supply chain [33]. The use of analytical methods is time-consuming, expensive, and requires personnel with specific skill sets [34]. The demand for more rapid, non-invasive, and sophisticated traceability systems in food supply chains is increasing so as to improve the standards of quality assurance [33]. Over the years, several advancements have been made in the field of vibrational spectroscopy, which has led to the development of portable instruments for tracing and monitoring the quality of food and feed where near-infrared (NIR) spectroscopy has been extensively used for this purpose [33,35,36]. NIR spectroscopy has been used to address the authenticity and traceability of food and feeds from two different aspects, including (1) the authenticity of production based on production parameters such as geographical origin, etc. [37–39], and (2) traceability based on the product description such as raw vs fresh, processed vs unprocessed, etc. [40–43]. Chemometric techniques such as principal components analysis (PCA), partial least squares (PLS), and principal component regression (PCR) are commonly used in combination with NIR spectroscopy in the traceability and authenticity of food and feeds [44–46]. The use of such chemometric techniques with NIR exhibits the potential to consider multiple components present within the system simultaneously and assist in obtaining information based on links with other sample characteristics [35,36].

This study aims to evaluate the ability of NIR spectroscopy to trace live fifth instar larvae and sixth instar BSF pre-pupae grown on organic side streams in an industrial setup. The combination of NIR spectroscopy with partial least squares discriminant analysis (PLS-DA) was used to develop classification models to predict the feed source used to grow the larvae.

## 2. Materials and Methods

### 2.1. Rearing of BSFL on Soy Waste and Spent Brewer's Grains

Homogenous soy waste and spent brewer's grain were used as two feed sources for rearing BSFL at a commercial BSFL production facility. Two kilograms of soy waste and spent brewer's grain was added separately to 6 trays (per feed source) of dimensions $60 \times 40$ cm as starter feed. Ten thousand 5-day old (5-DOL; chronological age) BSFL obtained upon hatching BSF eggs were added into each of the 6 trays containing the starter feed. The moisture content of the feed was maintained above 70% by measuring it using an A&D Weighing MF50 moisture analyser. All 6 trays containing the 5-Day Old (5-DOL) larvae were placed in the rearing room. The temperature of the rearing room was maintained above 25 °C, and the humidity was left ambient, monitored using moisture and temperature sensors. These conditions were set up by the production facility. The trays were monitored regularly and were fed ad libitum. The 5th instar larvae were collected upon the sighting of the first pre-pupae (6th instar) using sterile tweezers. Fifty grams of larvae were collected from 6 trays and pooled together, from which 50 live larvae were subjected to scanning by a NIR spectrometer. Randomness was ensured in sample collection by picking larvae from different spots in the trays. The pre-pupae (6th instars) were harvested in a similar way upon sighting the first pupae in the trays. Morphological criteria were used to identify the samples (5th and 6th instar).

### 2.2. Collection of NIR Spectra

The FT-NIR spectra of the larvae samples (5th and 6th instars) were collected using a Bruker Tango-R spectrophotometer (Bruker Optics GmbH, Ettlingen, Germany) with a gold-coated integrating sphere (diffuse reflection) (see Scheme 1). Samples (larvae samples from the 5th and 6th instars) were placed in a borosilicate-glass cuvette of 10 mm diameter (Bruker Optics GmbH, Ettlingen, Germany). The reflectance spectra of each sample were recorded using OPUS software (version 8.5, Bruker Optics GmbH, Ettlingen, Germany) with 64 interferograms at a resolution of $4 \text{ cm}^{-1}$ in the wavenumber range of 11,550 to $3950 \text{ cm}^{-1}$. Cuvettes were cleaned with 70% ethanol and dried with paper wipes between samples.

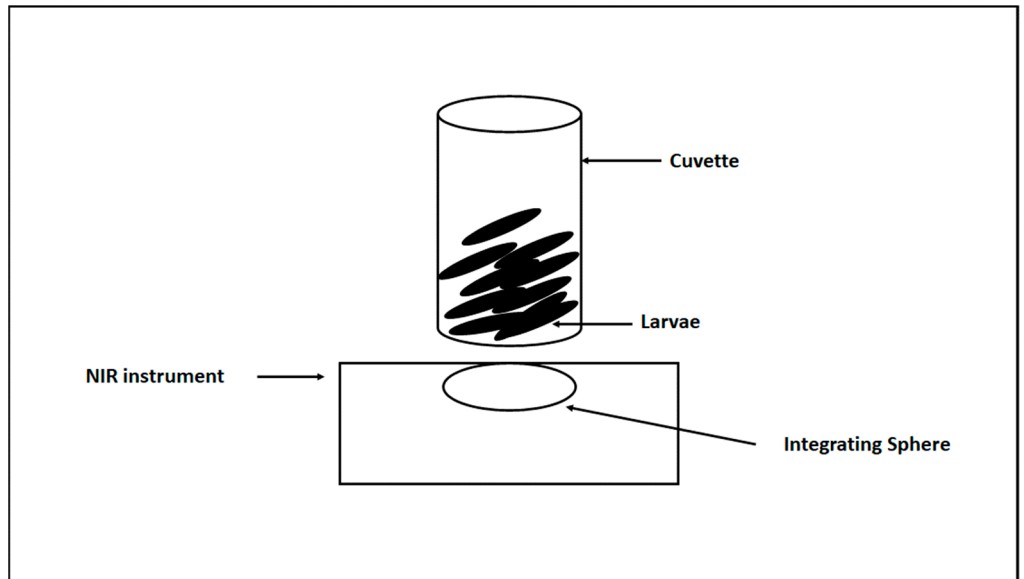

**Scheme 1.** Spectra collection of the larvae samples using near infrared spectroscopy.

### 2.3. Data Analysis

Multivariate analysis was performed using The Unscrambler X software (v11, CAMO ASA, Oslo, Norway). The NIR spectra were smoothened and pre-processed using the Savitzky–Golay second derivative (second order polynomial and a smoothing window size of 10 points) [47]. Principal component analysis (PCA) was performed to visualise the data structure and identify patterns, trends, outliers in the spectra, and other dominant features in the samples according to feed and instars [48,49]. Partial least squares regression discrimination models (PLS-DA) between the NIR spectra and reference data (dummy values for each feed) were developed using full cross-validation (leave one out) [50]. The optimal number of factors for the calibration model was selected based on the minimal value of the predicted residual sum of squares (PRESS), and the highest correlation coefficient ($R^2$) between actual and predicted values was used for the selection of the optimal number of factors for the calibration model being developed. The PLS models were evaluated using the standard error of cross-validation (SECV) and the coefficient of determination in cross-validation [50,51]. Linear discriminant analysis (LDA) was also used to classify the 5th and 6th instar samples based on the feed source used to rear the larvae.

## 3. Results and Discussion

### 3.1. NIR Spectra Interpretation

The second derivative of the NIR spectra of the BSFL samples analysed is displayed in Figure 1. It can be observed that the main variations in the NIR spectra are around 4320 and $4336 \text{ cm}^{-1}$, $4624 \text{ cm}^{-1}$, $5200 \text{ cm}^{-1}$, $5792 \text{ cm}^{-1}$, and 7296 and $7008 \text{ cm}^{-1}$. These bands are related at $5200 \text{ cm}^{-1}$ with O-H combination bands and O-H stretching in the 1st overtone region of water bands [52,53]. At $7008 \text{ cm}^{-1}$, this band is linked with $CH_2$ combinations associated either with proteins or carbohydrates, while at around $5792 \text{ cm}^{-1}$, this band

is associated with amine groups and proteins [54]. At around 4624 cm$^{-1}$ and 4336 cm$^{-1}$, these wavelengths are associated with both C-H aromatic groups and C-H combinations associated with lipids and compounds containing aromatic rings. Overall, it can be observed that common constituents present in BSFL, such as protein and lipids, contributed to explaining the observed variations in the NIR spectra of the samples analysed.

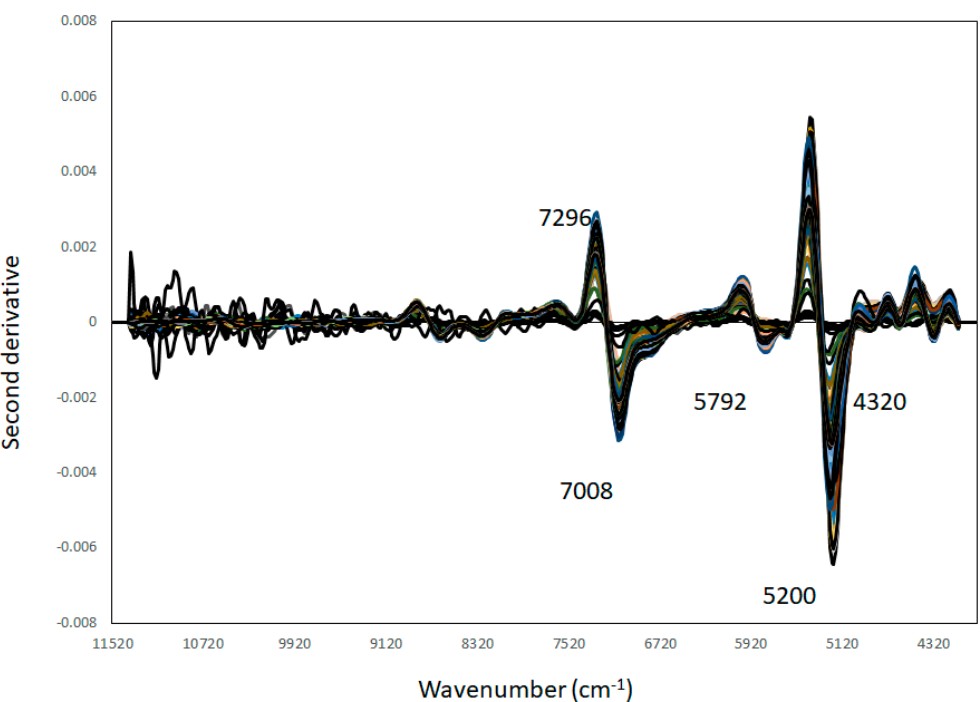

**Figure 1.** Second derivative NIR spectra obtained from 5th instar larvae and 6th instar pre-pupae reared on soy waste and spent brewer's grains.

### 3.2. Principal Component Analysis

Figures 2–4 show the PCA score plots obtained from the analysis of the NIR spectra of the fifth instars, sixth instar larvae, and pre-pupae reared on soy waste and spent brewer's grains. Figure 2 displays the PCA score plot obtained for analysis of larvae sourced from two different feeds. PC1 explains 71% of the variation in the samples, while 26% of the variation was accounted for by PC2. It can be observed that the fifth instar larvae and the sixth instar pre-pupae tend to cluster together and are clearly separated from one another. This suggests that the information collected in the NIR spectra is capable of differentiating the fifth instar BSFL larvae from the sixth instar pre-pupae irrespective of the feed source used to rear them. NIR spectroscopy has demonstrated its ability to identify the age of several insect species, including *Musca domestica* (common house flies), *M. autumnalis* (flies), *Stomoxys calcitrans* (flies), *Culicoides sonorensis* (midges), *Anopheles gambiae* (mosquito), and *Anopheles arabiensis* (mosquito) [55–58]. However, no reports were found on the use of NIR spectroscopy to distinguish BSFL samples based on their larval morphology. It can be interpreted from this PCA score plot (Figure 2) that the physiology of the larvae might play an important role in determining the traceability of BSFL. However, some fifth instar larvae were noted to overlap with the sixth instar pre-pupae samples—a plausible explanation is that some of the harvested fifth instar samples might be approaching the next morphological stage, which is the pre-pupae or the sixth instar. It was mentioned in the methods section that ten thousand 5-DOL (chronological age) larvae were introduced into the trays. However, there is a probability that certain eggs could have hatched earlier, resulting in the introduction of slightly older larvae than the intended (>5-DOL) larvae.

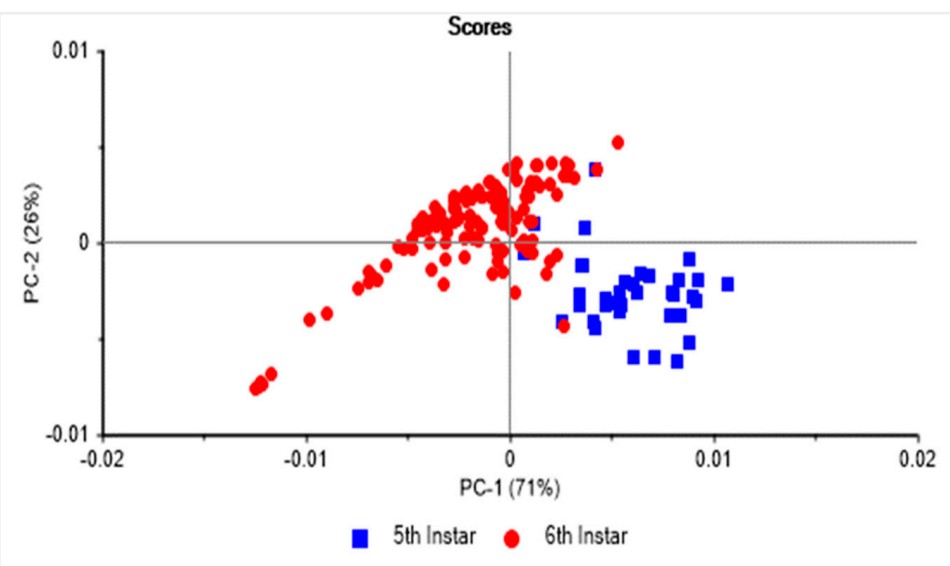

**Figure 2.** Principal component score plot of 5th instar larvae and 6th instar pre-pupae reared on soy waste and spent brewer's grains analysed using near infrared spectroscopy.

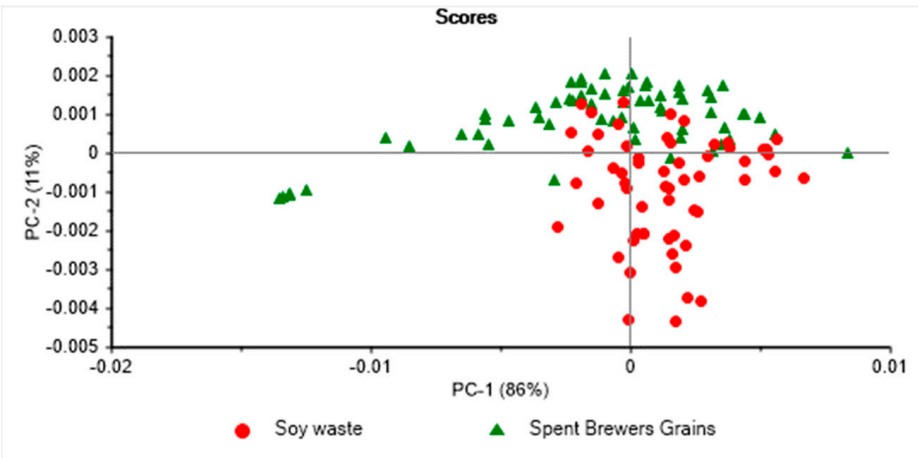

**Figure 3.** Principal component score plot of 5th instar larvae reared on soy waste and spent brewer's grains and analysed using near infrared spectroscopy.

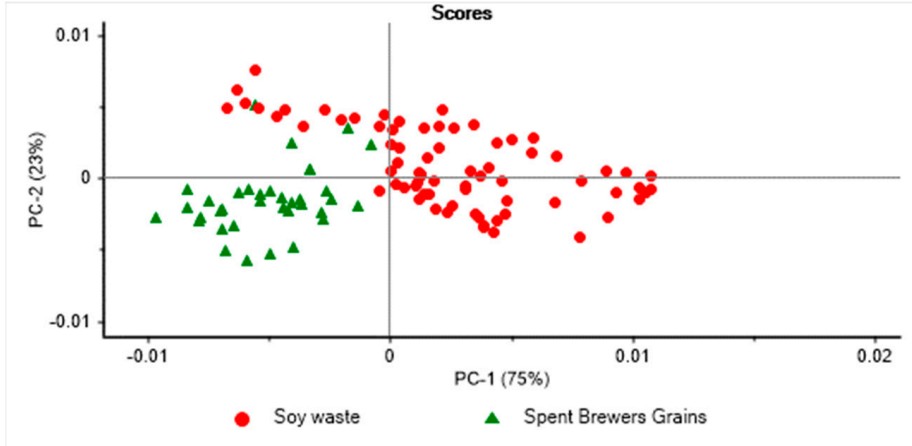

**Figure 4.** Principal component score plot of 6th instar pre-pupae reared on soy waste and spent brewers' grains and analysed using near infrared spectroscopy.

Figure 3 displays the PCA score plot of the fifth instar BSFL larvae reared on two different waste streams. PC1 explained 86% of the variation, while PC2 explained 11% of the variation in the samples analysed. The fifth instar BSFL reared on soy waste were found to form distinct clusters from those reared on spent brewer's grain. The PCA score plot for the sixth instar pre-pupae samples reared using two different waste streams is presented in Figure 4, where PC1 explained 75% of the variation, and PC2 accounted for 23% of the variation in the samples analysed. Figure 5 shows the loadings derived from the PCA analysis. It was observed that the PCA loadings were different for each of the PCA models developed. The highest PCA loadings were observed in most of the wavenumbers/wavelengths described in the above section.

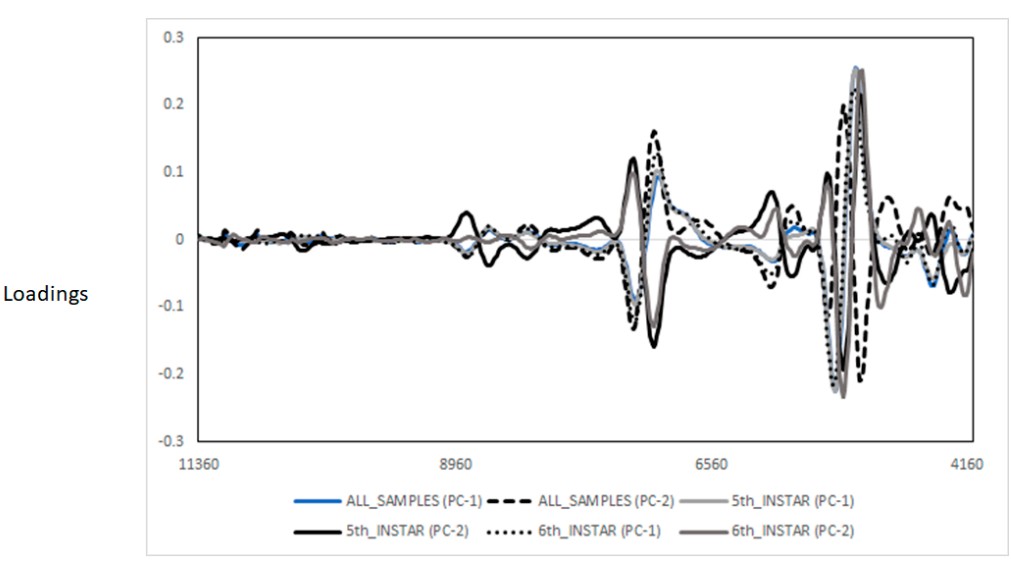

**Figure 5.** Loadings derived from the principal component analysis from the black soldier fly samples analysed using near infrared reflectance spectroscopy.

The results obtained in this study indicated that the sixth instar pre-pupae and fifth instar larvae tend to cluster as a result of the feed source used to rear them. Similar results were reported where powdered *Bombyx mori* was found to be separated into different groups based on the substrate used for rearing [59]. Similar trends were also observed when powdered crickets and buffalo worms obtained from two different commercial suppliers were analysed using mid-infrared spectroscopy [60]. Insects from the same species were separated into two different clusters due to the difference in their processing conditions [60].

### 3.3. PLS-Discriminant Analysis (DA)

Following the PCA analysis, PLS-DA classification was used to classify samples according to the feed source used to rear the BSFL instars (fifth instar larvae and sixth instar pre-pupae). The PLS-DA models were built separately for the two instars as PCA revealed that the physiological age of the larvae might play a role in the classification of larvae samples based on their feed source. The coefficient of determination ($R^2$) and standard error in cross-validation (SECV) obtained for the prediction of the feed source used to rear the fifth and sixth instar were 0.91 (SECV: 0.14) and 0.89 (SECV: 0.17), respectively. To identify the main regions of the NIR spectra that contribute to each of the models, the PLS loadings were interpreted. Figure 6 shows the optimal PLS loadings used for each of the PLS-DA models developed. It was observed that bands at 10,336 cm$^{-1}$, 7264 cm$^{-1}$, 5738 cm$^{-1}$, 5200 cm$^{-1}$, and 4900 cm$^{-1}$ contributed to explaining the PLS model obtained. The loadings used by the PLS model based on the fifth instar were higher than those observed for the sixth instar in the same wavenumbers, 10,192 cm$^{-1}$ (O-H), 7296 cm$^{-1}$ (O-H), 5840 cm$^{-1}$ (C-H and C-H$_2$), 5264 cm$^{-1}$ (C=O), and 4976 cm$^{-1}$ (N-H). In addition, the loadings were

opposite around 8704 cm$^{-1}$, 8320 cm$^{-1}$, and 4200 cm$^{-1}$ associated with C-H aromatic groups. Overall, these results indicated that the main source of variation between different feed sources and the two different instars are associated with differences in lipids (e.g., fatty acid profiles), protein content, and the presence of aromatic compounds.

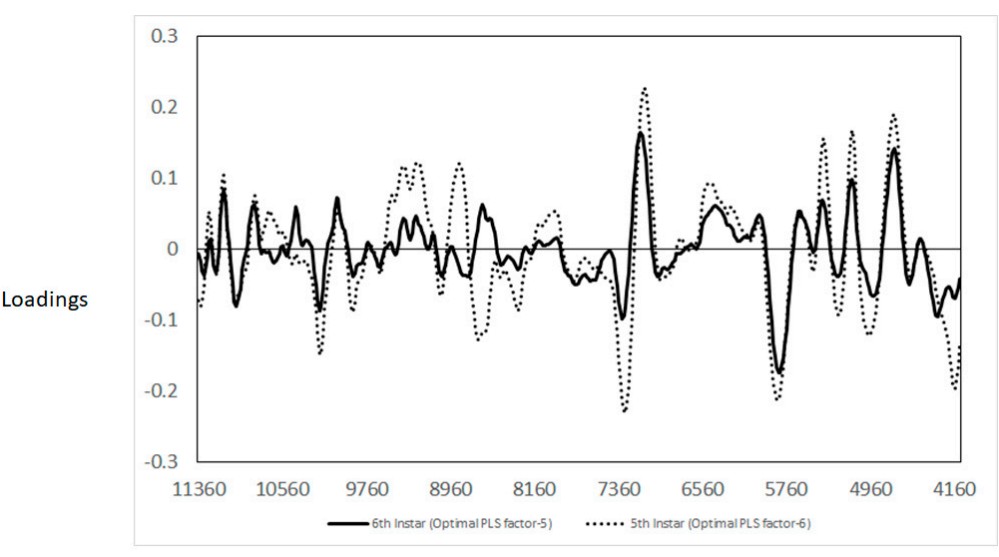

**Figure 6.** Partial least squares optimal loadings used to predict the source of stream feed utilised to rear the black soldier fly samples analysed using near infrared spectroscopy.

*3.4. Linear Discriminant Analysis*

Table 1 shows the LDA confusion matrix obtained for the classification of the feed source used to rear the fifth and sixth instar BSFL samples. A correct classification rate above 95% was obtained for the identification of the type of feed used to rear the larvae samples. The overall correct classification obtained was 97% and 96% using the fifth and sixth instars, respectively.

**Table 1.** Linear discriminant analysis (LDA) confusion matrix, percentage of correct classification. In brackets is the number of samples.

| Larval Development Stage | | Feed Source | |
| --- | --- | --- | --- |
| | | Spent Brewers' Grains | Soy Waste |
| 5th Instar | Spent brewers' grains | 100% (34/34) | 0 |
| | Soy waste | 3% (2/74) | 97% (72/74) |
| 6th Instar | Spent brewers' grains | 96% (64/67) | 4% (3/67) |
| | Soy waste | 0 | 100% (57/57) |

It is well-established that the larval composition is said to be highly influenced by the substrate used for rearing them and between different larval instars [61–63]. Based on this information, it is suggested that the difference in nutritional composition between the instars and feeds enabled the correct identification of the feed source used to rear the BSFL larvae.

## 4. Conclusions

The findings of the study suggested that NIR spectroscopy exhibits the potential to discriminate the morphology of BSFL (instars) as well as to trace the feed source used to grow the larvae. Further studies should be carried out to evaluate the ability of NIR spectroscopy to trace BSFL when reared on organic side streams such as food waste, municipal waste, abattoir waste and other heterogeneous matrices. Overall, NIR spectroscopy

exhibits the potential to be used as a non-invasive rapid tool for the real-time traceability of BSFL. A traceability system for BSFL will aid in risk evaluation and the identification of hazards associated with it, thereby assisting in improving the safety and quality of BSFL intended to be used by the animal feed industry. Although the results of the present study are promising, further research is still needed to validate the existing PLS-DA and LDA models, as well as to include chemical parameters associated with quality, such as crude protein and lipid content.

**Author Contributions:** S.A. and D.C.: sample preparation, analysis, draft preparation. L.C.H., S.M.O.M., D.M., P.J., O.Y., D.C. and S.A.: writing-review and editing of the manuscript. S.M.O.M., D.M., P.J., O.Y. and D.C.: supervision and project administration. L.C.H.: funding acquisition. All authors have read and agreed to the published version of the manuscript.

**Funding:** The work has been supported by the Fight Food Waste Cooperative Research Centre (Project 2.4.1), whose activities are funded by the Australian Government's Cooperative Research Centre Program.

**Institutional Review Board Statement:** Not applicable.

**Informed Consent Statement:** Not applicable.

**Data Availability Statement:** Not applicable.

**Acknowledgments:** The authors acknowledge Claire Nock, Viviane Filgueiras, and Alvin from Flyfarm for their help and support with sample preparation and collection.

**Conflicts of Interest:** The authors declare no conflict of interest.

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
