# Peer review of "Near Infrared Spectroscopy as a Traceability Tool to Monitor Black Soldier Fly Larvae (Hermetia illucens) Intended as Animal Feed"

_applsci, doi:10.3390/app12168168_

Round 1

Reviewer 1 Report

Review for the manuscript "Near infrared spectroscopy as a traceability tool to monitor black soldier fly larvae (Hermetia illucens) intended as animal feed", by S. Alagappan and co-authors.

The general problematic is well explained. The experimental design is clear, and the results clear to understand. I just have few concerns in the interpretation of the results (see remarks below). I recommend minor revisions, mostly in the presentation of the results, before final acceptance of the manuscript.

It was not clear to me how the transition from the living larva to the "sample" was made.  Is analysis performed directly on the larva, or after some preparation (dessication)?

* L169 5-DOL -> 5-day old ? 

* It could be interesting to provide a representation of the loadings of the PCA, in order to understand which parts of the spectrum allow to differentiate the instars or the feed sources.

* Figure 1 and 5: I suggest using a more regular tick labeling for the x-axis. That would also allow writing tick labels horizontally, thus facilitate reading

* Figure 2-3: the shape of the point cloud for the 6th instar in Figure 2 strangely looks like that of the whole point cloud in Figure 3... It looks like an interversion occured in the legend or in the processing workflow...

* Figure 5: it is quite confusing that the 6th instar is legended "factor-5", whereas the 5-th instar is legended "factor 6". It is also very error-prone!

* Figure 5: I would have appreciated a small labeling of some relevant peaks of the PLS loading. Maybe the peaks could be related to nutritional properties of the feed sources?

Author Response

We thanks the reviewer for their comments. Below and in bold the answers to the reviewers. 

It was not clear to me how the transition from the living larva to the "sample" was made.  Is analysis performed directly on the larva, or after some preparation (dessication)?  Line 147, highlighted in yellow. We have scanned the live larvae, we called the sample as well.

* L169 5-DOL -> 5-day old ? We have corrected as suggested by the reviewer.

* It could be interesting to provide a representation of the loadings of the PCA, in order to understand which parts of the spectrum allow to differentiate the instars or the feed sources.  We have added a new Figure (Figure 5) showing the PCA loadings as suggested by the reviewer.  It can be observed that each of the PCA models developed had different PCA loadings.

* Figure 1 and 5: I suggest using a more regular tick labeling for the x-axis. That would also allow writing tick labels horizontally, thus facilitate reading.  We have improved the axis as suggested by the reviewer.

* Figure 2-3: the shape of the point cloud for the 6th instar in Figure 2 strangely looks like that of the whole point cloud in Figure 3... It looks like an interversion occured in the legend or in the processing workflow...We do not understand the comment from the reviewer.  The PCA reported in figure 2 was done using all samples combined the 6th instar and 5 day old larvae.  In the case of Figure 3 only samples from the 5th instar were used while in Figure 4 samples from the 6th instar.  The amount of explained variance is different and in the new figure added (new Figure 5) the PCA loadings are very different.

* Figure 5: it is quite confusing that the 6th instar is legended "factor-5", whereas the 5-th instar is legended "factor 6". It is also very error-prone!  The factor refers to the optimal number of terms or factors used by the PLS-DA model.  Nothing to do with the sample used.  It seems that the reviewer is picking up in random and not significant details.

* Figure 5: I would have appreciated a small labeling of some relevant peaks of the PLS loading. Maybe the peaks could be related to nutritional properties of the feed sources? We have improved the Figure as suggested by the reviewer.

Reviewer 2 Report

1. Excessive citation in line 103. It is necessary to provide a more detailed description of the sources [37-43].

2. The process of measuring spectra should be described in more detail in Section 2.2. Without this, including photographs, diagrams, it is difficult to judge the quality of the experiments performed.

3. In Figure 1, the numbers along the abscissa axis must be presented in a generally accepted format with equal intervals. The same remark is made in Figure 5.

4. It is not clear from Figure 5 which spectrum refers to which object.

5. The "Conclusion" section should be improved.

6. The manuscript does not propose a specific algorithm for monitoring using NIR spectroscopy.

Author Response

Please find the answers to the reviewers below.

  1. Excessive citation in line 103. It is necessary to provide a more detailed description of the sources [37-43]. We do not consider that using 6 references is excessive. These references are related with recent reviews on the use of vibrational spectroscopy to trace and authenticate foods.

  1. The process of measuring spectra should be described in more detail in Section 2.2. Without this, including photographs, diagrams, it is difficult to judge the quality of the experiments performed. We do not understand the statement by the reviewer that is difficult to judge the quality of the experiments. It is not much to add about the spectra collection.  However, we have added a diagram showing how the spectra of the samples was collected as suggested by the reviewer.

  1. In Figure 1, the numbers along the abscissa axis must be presented in a generally accepted format with equal intervals. The same remark is made in Figure 5. We have improved the axis as suggested by the reviewer.

  1. It is not clear from Figure 5 which spectrum refers to which object. The Figure 5 (now Figure 6) does not refer to any spectrum. The figure reports the optimal loadings used by the PLS regression to predict the feed source. We have improved the Figure.

  1. The "Conclusion" section should be improved. We have improved the conclusion as suggested by the reviewer.

  1. The manuscript does not propose a specific algorithm for monitoring using NIR spectroscopy. We cannot understand this comment from the reviewer. We have reported a feasibility study where we have used either PLS-DA or LDA as classification method.  These two algorithms are one of the most used in applications of NIR spectroscopy.   

Reviewer 3 Report

The manuscript titled “Near infrared spectroscopy as a traceability tool to monitor black soldier fly larvae (Hermetia illucens) intended as animal feed” deals with the ability of near infrared (NIR) spectroscopy for traceability applications in the food waste used to grow the BSFL.

The research covers a current topic, especially for the great potential of BSFL in the waste valorisation process though the bioconversion of different organic waste to produce valuable larval biomass and useful by-products in the context of a circular economy.

Findings are interesting also for a possible commercial application in the perspective of assessing safety issues of BSFL for animal feed.

The manuscript is clear and well written, but there is still a margin of improvement.

The Introduction section offers a wide background on relevant aspects of the bioconversion process by Hermetia illucens larvae, pointing up the role in the waste valorization to produce larvae as feed ingredient. A wide paragraph is dedicated to BSF life cycle, BSFL larval development and their morphological features with several citations. However, the fine descriptions of these different larval instars still neglect to emphasize that larval developmental stages are characterized by different nutritional and physiological properties: this aspect is the basis to plan the experimental design of this work and it is not just a conclusion or the interpretation of the NIR spectroscopy and further outcomes. “It is noteworthy that the morphological features of different larval instars are influenced by the substrate and rearing conditions” (lines 69-70) should point the importance of physiological and nutritional features, too. Some relevant studies reported these issues with particular emphasis on the physiological and nutritional value of BSF 5th instar and prepupae (6th) reared on organic wastes to produce valuable biomass (Liu et al., 2017; Giannetto et al., 2020a, b).

For these purposes, I suggest the authors to include these references:

-Veldkamp, T., Bosch, G., 2015. Insects – a protein rich feed ingredient in pig and poultry

diets. Anim. Front. 5, 45–50. https://doi.org/10.2527/af.2015-0019.

- Giannetto A, Oliva S, Riolo K, Savastano D, Parrino V, Cappello T, Maisano M, Fasulo S, Mauceri A. 2020. Waste valorization via Hermetia illucens to produce protein-rich biomass for feed: insight into the critical nutrient taurine. Animals 10: 1710. https://doi.org/ 10.3390/ani10091710.

- Giannetto, A., Oliva, S., Ceccon Lanes, C.F., de Araújo Pedron, F., Savastano, D., Baviera, C., Parrino, V., Lo Paro, G., Spanò, N., Cappello, T., Maisano, M., Mauceri, A., Fasulo, S., 2020. Hermetia illucens (Diptera: Stratiomydae) larvae and prepupae: Biomass production, fatty acid profile and expression of key genes involved in lipid metabolism. J. Biotechnol. 307, 44-54. http://doi.org/10.1016/j.jbiotec.2019.10.015.

The experimental design is simple but some details are missing; did the authors use these morphological criteria to distinguish 5th and 6th instars? Do they evaluate the criteria by Kim et al.? Although the goal of the study was to evaluate the ability of using NIR spectroscopy for tracing 5th and 6th instar BSFL grown on organic side streams, the rearing condition are not clear “The temperature of the rearing room was maintained above 25ºC and the humidity was left ambient” (lines 143-144).

In the Results and Discussion section, the authors report on “morphological” features, but the physiological peculiarities due to the differences in protein and lipid contents of the two developmental stages (not “morphological stage”, line 206) should be emphasized once again, as described by the suggested references.

The outcomes give new tools to assess the nutritional and safety issues in the perspective of using BSFL as feed ingredient in the next future.

Minor points

The name of the species is not always italicized. Please check throughout the text and reference list (i.e., 4, 5, 9, 12, 14, 15, 16, 18, 19, 21, 24, etc…).

Please check the space in the reference list.

doi is not reported in the same format; please check the author guidelines.

Table 1, p.10 is not bold; please check the author guidelines.

Line 200: please fix “An. Arabiensis”;

Line 220: check for double spaces here and throughout the text.

Author Response

“It is noteworthy that the morphological features of different larval instars are influenced by the substrate and rearing conditions” (lines 69-70) should point the importance of physiological and nutritional features, too. Some relevant studies reported these issues with particular emphasis on the physiological and nutritional value of BSF 5th instar and prepupae (6th) reared on organic wastes to produce valuable biomass (Liu et al., 2017; Giannetto et al., 2020a, b).  Thank you for the comment. I hope these are not the reviewers’ own references as I will found this of a completely lack of ethics and professionalism.

-Veldkamp, T., Bosch, G., 2015. Insects – a protein rich feed ingredient in pig and poultry diets. Anim. Front. 5, 45–50. https://doi.org/10.2527/af.2015-0019.

- Giannetto A, Oliva S, Riolo K, Savastano D, Parrino V, Cappello T, Maisano M, Fasulo S, Mauceri A. 2020. Waste valorization via Hermetia illucens to produce protein-rich biomass for feed: insight into the critical nutrient taurine. Animals 10: 1710. https://doi.org/ 10.3390/ani10091710.

- Giannetto, A., Oliva, S., Ceccon Lanes, C.F., de Araújo Pedron, F., Savastano, D., Baviera, C., Parrino, V., Lo Paro, G., Spanò, N., Cappello, T., Maisano, M., Mauceri, A., Fasulo, S., 2020. Hermetia illucens (Diptera: Stratiomydae) larvae and prepupae: Biomass production, fatty acid profile and expression of key genes involved in lipid metabolism. J. Biotechnol. 307, 44-54. http://doi.org/10.1016/j.jbiotec.2019.10.015.

The experimental design is simple but some details are missing; did the authors use these morphological criteria to distinguish 5th and 6th instars? Yes. This is explained in the materials and methods section.

Do they evaluate the criteria by Kim et al.? Although the goal of the study was to evaluate the ability of using NIR spectroscopy for tracing 5th and 6th instar BSFL grown on organic side streams, the rearing condition are not clear “The temperature of the rearing room was maintained above 25ºC and the humidity was left ambient” (lines 143-144).  Please note that the samples were obtained/collected from an industrial site.  These conditions were defined and used by the industrial site.

In the Results and Discussion section, the authors report on “morphological” features, but the physiological peculiarities due to the differences in protein and lipid contents of the two developmental stages (not “morphological stage”, line 206) should be emphasized once again, as described by the suggested references.  We do not understand the reviewer point.

Minor points

The name of the species is not always italicized. Please check throughout the text and reference list (i.e., 4, 5, 9, 12, 14, 15, 16, 18, 19, 21, 24, etc…). We have fixed as suggested by the reviewer.

Please check the space in the reference list. We have fixed as suggested by the reviewer.

doi is not reported in the same format; please check the author guidelines.

Table 1, p.10 is not bold; please check the author guidelines. We have fixed as suggested by the reviewer.

Line 200: please fix “An. Arabiensis”;  We have fixed as suggested by the reviewer.

Line 220: check for double spaces here and throughout the text.  Thank you for your comment. The doubles spaces were checked and fixed.

Round 2

Reviewer 2 Report

Dear authors!

I am extremely annoyed that most of my comments caused you to misunderstand. The comments were dictated by the need to improve the manuscript for a better understanding by Readers of the journal "Applied Sciences". I hope for your understanding.

Remark 1: In the citation [37-43] there are not six references, but seven! In my comment, it was recommended to describe them in a little more detail: give one or two phrases to each link (or two or three links). This is necessary for the convenience of understanding by Readers.

Remark 2: A more detailed description of the experiments is necessary for their possible reproduction by other researchers. What is the repeatability of measurements? What is the margin of error?

Section 2.2. is entitled "Collection of NIR spectra and Data Analysis", but the data analysis is given in Section 2.3.

Remark 3: Figure 1 is of very poor quality: the numbers on the axes are practically invisible. The range of values along the axes should be a multiple of 1, 2, 5 or 25. (This applies to Figures 1, 5 and 6).

Remark 4: It is still unclear what kind of set of unsigned curves is shown in Figure 1? What do the authors want to show readers in this drawing?

Remark 6: The title of the article "Near infrared spectroscopy ..." suggests a description of near infrared spectroscopy as a monitoring tool for Hermetia illucens larvae. If the authors believe that the main purpose of the article is "feasibility study where we have used either PLS-DA or LDA as classification method", then the title of the article should be corrected.

Author Response

We are very sorry to annoy the reviewer.  This was not our intention. We believe in a frank and professional review process.  However, the attitude of the reviewer is not very professional.  It seems the reviewer does not understand the paper, the technology and the data analysis.  We are sorry about this.

Remark 1: In the citation [37-43] there are not six references, but seven! In my comment, it was recommended to describe them in a little more detail: give one or two phrases to each link (or two or three links). This is necessary for the convenience of understanding by Readers.  We have corrected as suggested.

Remark 2: A more detailed description of the experiments is necessary for their possible reproduction by other researchers. What is the repeatability of measurements? What is the margin of error?  I will suggest the reviewer to read the vast number of references in the field.  The experiments are well detailed as per the NIR spectra collection, sampling and samples.  The repeatability and margin of error are related with the signal to noise ratio of the instrument.  We are using commercial available instrumentation, already prove.

Section 2.2. is entitled "Collection of NIR spectra and Data Analysis", but the data analysis is given in Section 2.3.  In order to avoid annoying the reviewer, we have edited and corrected the title of the section.

Remark 3: Figure 1 is of very poor quality: the numbers on the axes are practically invisible. The range of values along the axes should be a multiple of 1, 2, 5 or 25. (This applies to Figures 1, 5 and 6). We have modified the range.

Remark 4: It is still unclear what kind of set of unsigned curves is shown in Figure 1? What do the authors want to show readers in this drawing?  Once again, we suggest that the reviewer look at the vast number of literature in the field. We also suggest the reviewer to read some of our papers, some of them published in this journal.  Figure 1 shows the NIR spectra.

Remark 6: The title of the article "Near infrared spectroscopy ..." suggests a description of near infrared spectroscopy as a monitoring tool for Hermetia illucens larvae. If the authors believe that the main purpose of the article is "feasibility study where we have used either PLS-DA or LDA as classification method", then the title of the article should be corrected. This is out of context.  I will suggest the reviewer to be more professional in his/her comments.